# Interstitial Lung Abnormalities: Unraveling the Journey from Incidental Discovery to Clinical Significance

**DOI:** 10.3390/diagnostics15040509

**Published:** 2025-02-19

**Authors:** Athena Gogali, Christos Kyriakopoulos, Konstantinos Kostikas

**Affiliations:** Respiratory Medicine Department, University of Ioannina, Stavrou Niarchou Avenue, 45500 Ioannina, Greece; ckyriako@yahoo.gr (C.K.); ktkostikas@gmail.com (K.K.)

**Keywords:** interstitial lung abnormalities, pulmonary fibrosis, lung cancer, COPD, high-resolution computed tomography, mortality

## Abstract

Interstitial lung abnormalities (ILAs) are incidental radiologic abnormalities on chest computed tomography (CT) examination performed on people in whom interstitial lung disease (ILD) is not suspected. Despite the fact that most of these individuals are asymptomatic, ILAs are not synonymous with subclinical ILD, as a subset of them have symptoms and lung function impairment. On the other hand, not all ILAs progress to clinically significant ILD. Specific imaging features and patterns have been proven more likely to progress, while some individuals may comprise a higher risk group for progression. Numerous studies have demonstrated that ILAs are not only associated with an increased risk of progression toward pulmonary fibrosis and fibrosis-related mortality but are also linked to a greater incidence of lung cancer and a higher rate of all-cause mortality. Considering that the systematic evaluation of large cohorts has shown a prevalence of ILAs up to 7% and that the natural history of ILAs is unclear, successful screening and appropriate monitoring of ILAs is of particular significance for earlier diagnosis, risk factor modification, and treatment. The present review aims to summarize the current knowledge on ILAs and highlight the need to define those at greatest risk of progression to ILD and worse clinical outcomes.

## 1. Introduction

The increasing prevalence of interstitial lung disease (ILD), especially idiopathic pulmonary fibrosis (IPF), a cardinal fibrotic lung disease, accounts for significant morbidity and mortality worldwide. Other types of ILD express a progressive phenotype as well, with self-sustaining fibrosis, which leads to early mortality [1]. In the antifibrotic therapy era, especially with studies demonstrating that antifibrotic therapies are beneficial also in patients with more preserved lung function [2,3], the inevitably progressive nature of pulmonary fibrosis prompts early diagnosis and appropriate intervention, with improved life expectancy if treated. High-resolution computed tomography (HRCT) plays a central role in the assessment of patients with ILD and is often diagnostic.

The increasing use of computed tomography (CT) imaging and improving quality have remarkably increased the possibility of ILD being an incidental CT finding. Interstitial lung abnormality (ILA) is a radiological term that captures this incidental discovery of early ILD. Lately, lung cancer screening has provided a unique opportunity for the identification of ILAs [4]. The clinical importance of ILAs is increasingly recognized, as a significant subset of individuals have symptoms and physiological impairment [5]. In many instances, there is a strong association between ILAs and disease progression and ILD-related, cancer-related, and all-cause mortality [6,7]. Despite their prognostic significance, the recognition of subtle interstitial abnormalities may be rather challenging, and there is an increased risk of remaining unreported in almost half of the cases [8]. An additional problem is that ILAs are relatively frequent, especially in older adults [9], but only a minority will progress into clinically significant disease; moreover, in the absence of follow-up guidelines, the management of such patients represents a clinical challenge.

The present review aims to summarize the most up-to-date knowledge on ILAs and highlight the need to identify patients at the greatest risk of progression to ILD and worse clinical outcomes, with the final goal of eliminating potential risk factors and offering appropriate early treatment. Besides contributing to disease awareness, we aimed to comment in detail on issues not adequately addressed, such as the association between ILAs and genetic predispositions, cancer, and COPD, and also provide a simple framework for the management of ILAs for the practicing clinician.

## 2. What Are ILAs?

### 2.1. Definition

A multidisciplinary Position Paper by the Fleischner Society in 2020 provided standardization of the definition and terminology of ILAs [10]. ILAs is a purely radiological term based on the incidental identification of non-dependent abnormalities, recognized on thoracic or abdominal CT scans performed for various clinical indications except clinical suspicion of ILD. These include ground-glass or reticular abnormalities, lung distortion, traction bronchiectasis/bronchiolectasis, honeycombing, and non-emphysematous cysts. The abnormalities involve at least 5% of a lung zone (upper, middle, and lower lung zones are demarcated by the levels of the inferior aortic arch and right inferior pulmonary vein) and are denoted in individuals in whom interstitial lung disease is not suspected. Thus, by definition, abnormalities found in radiological imaging of screening high-risk populations, such as relatives of patients with known ILD or patients with connective tissue disease (CTD) and individuals with occupational exposure, are not considered ILAs but rather preclinical or subclinical ILD. It is important to stress that ILAs are not synonymous with subclinical ILD because a subset of patients with ILAs have symptoms not previously attributed to an ILD condition and abnormal lung function. ILAs fall into the mild ILD category when an individual is reported with ILA findings and then clinical investigation reveals symptoms, clinical signs, or functional impairment. Imaging findings restricted to dependent lung atelectasis, focal paraspinal fibrosis in close contact with thoracic spine osteophytes, mild focal or unilateral abnormality, interstitial oedema (e.g., in heart failure), pleuropulmonary fibroelastosis elements (a well-defined entity), or findings of aspiration or smoking-related centrilobular nodularity in the absence of other findings (pathology not related to progression or fibrosis and thus not clinically significant) are not considered ILAs and should not provoke confusion to clinicians (Table 1).

### 2.2. ILA Subcategorization

ILAs are subcategorized radiologically [10] based on prognostic significance implications in the following:Non-subpleural ILAs: ground-glass opacity and reticular opacities without a predominant subpleural localization.Subpleural nonfibrotic ILAs: ground glass opacity and reticular opacities with a predominant subpleural localization and without evidence of fibrosis.Subpleural fibrotic ILAs: with a predominant subpleural localization and with evidence of pulmonary fibrosis (traction bronchiectasis, architectural distortion, and honeycombing).

ILAs with a non-subpleural distribution are usually non-progressive and do not have survival implications, while subpleural ILAs, in general, are potentially clinically significant. Fibrotic ILAs are clearly associated with higher odds of ILA progression and increased mortality [11] (Figure 1).

## 3. How Prevalent Are ILAs?

ILAs are more prevalent in older adults over 60 years old and in smokers. In large population-based cohorts such as the Framingham Heart Study (FHS) and the Age Gene/Environment Susceptibility (AGES)-Reykjavik study, the prevalence of ILAs varies from 3% to 10%, with a mean age from 62 to 78 years old [6,7], while in smoking and lung cancer screening cohorts such as the Genetic Epidemiology of COPD Study (COPDGene) [12] and the Evaluation of COPD Longitudinally to Identify Predictive Surrogate End-points (ECLIPSE), the mean age was approximately 10 years younger, ranging from 62 to 64 years old, while prevalence ranged from 4% to 17% [6,7]. The estimated prevalence is 4–9% in cigarette smokers and 2–7% in never-smokers [10]. Pesonen et al. underline the fact that never-smokers are a population that is at risk of underdiagnosis, as although they do not undergo lung cancer screening, ILAs are common, especially the nonfibrotic phenotype [13]. In this population, ILAs are associated with older age, low-grade inflammation, a restrictive spirometric pattern, and impaired diffusing capacity [13].

The prevalence of ILAs in high-risk populations, as in first-degree relatives of patients with fibrosis [14] or with known CTD [15], is much higher, up to 30% and 22%, respectively, and findings should be considered as preclinical ILD most likely to progress.

## 4. What Is the Histopathology Behind ILAs?

Lung tissue specimens obtained by pulmonary resection for lung cancer or lung nodule revealed that the majority of the patients with ILAs had smoking-related interstitial fibrosis, but usual interstitial pneumonia (UIP), pulmonary Langerhans’ cell histiocytosis, nonspecific interstitial pneumonia, and asbestosis were also identified [16]. It is interesting that CT scans underestimate the incidence of pathological ILD, as imaging resolution could not capture histological fibrosis in 52% of individuals characterized with no ILAs or intermediate radiological changes [17,18]. The fact that most of the studies show that ILAs represent smoking-related changes and quite less frequently UIP may not be quite accurate in the general population, as histological specimens are derived mostly from heavy smokers. In a blinded, systematic comparison of radiological imaging and histopathology in a lung nodule resection cohort, subpleural fibrosis, fibroblastic foci, and atypical adenomatous hyperplasia were the findings strongly related to ILA [17].

## 5. What Are the Risk Factors for ILAs?

### 5.1. Demographics

It is interesting that ILAs share common risk factors with IPF. Increasing age is definitely an important risk factor for ILA, as shown in most studies [19]. In the COPDGene study, the prevalence of ILA increased remarkably in those aged 70 years or older (4% in patients <60 years old to 6% in those >70 years old) [12], while in the Framingham Heart Study, the prevalence of ILA in those aged 70 or older was up to 47% [6,7]. The presence of ILA was associated with increased levels of biomarkers of aging and has sometimes been considered as part of the normal aging of the lungs [20]. However, the observations that the progression of ILA is more likely in advanced age and that ILA is an independent-of-age factor of increased mortality suggest that ILA is an important comorbidity in older patients. Male sex was identified as a risk factor in some studies, but this was not a consistent finding in all studies [21].

### 5.2. Exposures

Occupational exposures to vapors, gases, dust, and fumes increase ILA risk [22], with tobacco smoke exposure being the most significant risk factor (1.8 times increase). Cumulative and current cigarette smoking effects on ILAs found in the Multi-Ethnic Study of Atherosclerosis (MESA) population demonstrated that the prevalence of spirometric restriction increased relatively by 8% for every 10 pack-years, and the volume of high-attenuation areas increased by 1.6 cm³ for every 10 pack-years [23]. The specific ILA mostly associated with current smoking is the pattern of centrilobular nodules (odds ratio [OR] = 4.82) [12]. In a cohort of 352 patients that underwent resection of lung nodules or masses, those with radiographic ILAs/ILDs were more frequently ever-smokers (49% vs. 39.9% with no ILAs), with more pack years of smoking (44.57 ± 36.21 vs. 34.96 ± 26.22) and more often with concurrent COPD (35% vs. 26.5%) and radiographic evidence of centrilobular (40% vs. 22.2%) and paraseptal emphysema (17.6% vs. 9.6%) [24].

Air pollution also merits special attention, as it seems to be involved in pathogenetic pathways such as oxidative stress, inflammation, and telomere shortening [25]. Sack et al. [26] performed an analysis of 6,813 individuals enrolled in the MESA study followed for 10 years with serial CTs, and predictions of ambient pollution at each home were made. ILA risk increased 1.62-fold per 40 ppb increment in nitric oxide (NOx) and was strongest in non-smokers (2.60-fold increase per 40 ppb increment in NOx). High-attenuation abnormalities increased by 0.54% per year per 5 μg/m³ increment in fine particulate matter (PM2.5) and by 0.55% per year per 40 ppb increment in NOx. Rice et al. [27] evaluated associations of ILA with long-term exposure to traffic and ambient pollutants. Exposure to PM2.5, elemental carbon (EC, a traffic-related PM2.5 constituent), and ozone was examined. An IQR difference in 5-year EC exposure of 0.14 μg/m3 was linked with a 27% greater probability of ILA and a 33% greater probability of ILA progression, while associations of the other measures of ambient pollution were inconclusive. By contrast, air pollution exposure was not an independent risk factor for ILA occurrence in the SubPopulations and InteRmediate Outcome Measures In COPD Study (SPIROMICS) population with COPD. The association between pollution and ILA was affected by the presence of genetic polymorphisms (relative risk [RR] per 26 ppb NO_x_, 2.41 and RR per 4 μg ⋅ m^−3^ PM_2.5_, 1.43, for curriers) and the history of smoking habit (RR per 10 ppb, NO_2_ 1.64 for former smokers) [28].

### 5.3. Genetics

It is well known that the promoter polymorphism rs35705950 in the gene encoding mucin 5B (*MUC5B*) is associated with IPF and thus is the most commonly investigated in ILAs. In participants in the Framingham Heart Study, the relationship between the ILAs and this genotype was found to increase the risk for ILA by 2.8 times, and the odds of definite CT evidence of pulmonary fibrosis were 6.3 times greater. Moreover, the association was greater in older adults, and there was no link to smoking [29]. Furthermore, for each copy of the minor allele of the *MUC5B* promoter polymorphism, there was a 0.64% absolute increase in the percentage of the lung with interstitial changes [30], the *MUC5B* genotype was strongly associated with the subpleural subtype of ILA and was predictive of a UIP pattern in non-Hispanic white populations [31]. Common genetic variants outside the *MUC5B* region can also be associated with an increased risk of ILA, ILA progression, and IPF. In a study of 14,650 participants, polygenic risk scores with and without the *MUC5B* region on IPF, ILA, and ILA progression were developed. Polygenic risk scores without the *MUC5B* region showed an almost equal association with IPF with the *MUC5B* variant. They were also associated with ILA and ILA progression, and they reinforced the *MUC5B* variant in the identification of high-risk individuals for interstitial lung abnormalities and pulmonary fibrosis [32]. A genome-wide association study that included 1,699 individuals with ILAs and 10,274 control subjects from 6 cohorts showed that ILA shares common genetic pathways with IPF but also separate ones. The *MUC5B* promoter variant rs35705950 and five more previously reported IPF loci were significantly associated with both ILAs and subpleural ILAs. Additionally, novel genome-wide associations near *IPO11* (rs6886640) and (rs73199442) with ILAs and near *HTRE1* (rs7744971) with subpleural-predominant ILAs were discovered, unrelated to IPF. This is indicative that some ILA represents an early phase of other types of ILD, distinct from IPF [33].

Putman et al. showed that ILAs are associated with decreased mean telomere length (MTL). In the COPDGene and AGES-Reykjavik cohorts, there was a greater than twofold risk for ILA when comparing the shortest quartile of telomere length to the longest quartile, while in the FHS, those with ILA had shorter telomeres compared to those without ILA [34]. A comparison of airway gene expression in those with ILAs with a UIP pattern with those with no ILA revealed 17 different pathways in a population of smokers [35]. Proteomics may help to identify patients at risk for pulmonary fibrosis. The proteomic profile of ILAs showed that surfactant protein B, WAP four-disulfide core domain protein 2, and Secretoglobin family 3A member 1 were significantly associated with ILA, and the first two correlated strongly with ILA progression [36].

The clinical implications of genetic factors in ILA are more obvious in relatives of familial pulmonary fibrosis patients, in whom ILAs on chest CT scans are common, with a reported prevalence in first-degree relatives of 14–25%. Among them, 63.3% of patients with limited ILAs at baseline experienced progression, while 19.4% developed extensive HRCT abnormalities or clinical ILD at 5 years [37].

Soluble biomarkers have been investigated also in ILAs. Buendía-Roldán et al., in a lung aging program at the Mexican National Institute of Respiratory Diseases, found that approximately 10% of asymptomatic individuals enrolled had ILAs [38]. The serum levels of resistin, matrix metalloproteinase (MMP)-1, MMP-7, MMP-13, interleukin (IL)-6, and surfactant protein (SP)-D significantly correlated with the presence of ILAs; resistin and MMP-13 levels were the most markedly increased [38]. Recently, Sanders et al. found a connection between ILA and plasma biomarkers of accelerated aging [20]. In the FHS, increasing plasma concentration of growth differentiation factor 15 was strongly associated with the presence of ILA (OR = 3.4), TNFR (OR = 3.1), IL-6 (OR = 1.8), and CRP (OR = 1.7). Associations of GDF15 with mortality approached significance. In the COPDGene Study, GDF15 was strongly associated with ILA (OR = 8.1) and mortality as well (hazard ratio [HR] 1.6). Causal inference analysis showed that the association of age with ILA was mediated robustly by GDF15 in both cohorts. These results imply that aging and the development of ILA share common pathogenic pathways.

Finally, it is known that higher blood monocyte counts are associated with shorter survival in patients with pulmonary fibrosis. It has been shown that they also are associated with the presence and progression of interstitial lung abnormalities as well as with lower forced vital capacity (FVC) [39].

## 6. What Is the Clinical Importance of ILAs?

### 6.1. Presence of Respiratory Symptoms and Physiological Impairment

As mentioned previously, ILA diagnosis is not synonymous with asymptomatic disease. Participants with ILAs in FHS were more likely to have shortness of breath, chronic cough, and reduced total lung and diffusion capacity [29]. The associations between ILA and self-reported measures of health and functional status in participants from the AGES-Reykjavik study were explored. Participants with ILA were 30% less likely to be independent in activities of daily living, had a 34% lower chance to have good or better self-reported health status, and had a 28% lower chance to participate in physical activities [40]. Similarly, people with ILAs in the COPDGene cohort, both with and without COPD, had a reduced 6-min walk test distance than patients without ILAs [41]. Finally, it was recently shown that suspected ILD was already present in half of the subjects with ILA in COPDGene and was associated with reduced exercise capacity, increased symptoms, supplemental oxygen need, and severe respiratory exacerbations [42].

### 6.2. Progression to ILD and Respiratory Related Mortality

Individuals with ILAs have an increased risk of developing ILD with subsequent increased risk of respiratory morbidity and mortality. In a US population cohort, high-attenuation areas were associated with an increased rate of ILD hospitalization (HR = 2.6 per 1-SD increment in high-attenuation areas) and an increased rate of ILD-specific death (HR = 2.3), both in smokers and non-smokers [43]. It is estimated that 20% of ILA cases progress at 2 years and 46% at 4 years [10]. The radiological subtype is of major importance for progression risk [6]. ILAs were assessed in 1,867 participants in FHS who had serial CT scans in a 6-year period. A total of 21% showed definite progression, 55% had probably progressed, and only 15% had shown stability. Of those who developed progressive imaging abnormalities, 91% ultimately had definite fibrosis, of whom one-tenth had developed a UIP pattern. Those with ILA progression were older, had increasing copies of the *MUC5B* promoter polymorphism, and experienced an accelerated decline of FVC. However, the annual decline in FVC in patients with ILA progression was approximately 64 mL, less than the annual decline in FVC generally noted among patients with IPF in randomized controlled trials (approximately 200 mL per year). Overall, participants without ILA had a mean decrease in FVC of 35 mL per year, while those with ILA without progression had a mean decrease of 40 mL per year. Those with progressive imaging abnormalities had an increased risk for death (HR = 3.9) after 4 years of follow-up. Putman et al. [11] reported that specific imaging patterns are associated with ILA progression and mortality. In the AGES-Reykjavik study, 73% of participants had imaging progression over 5 years linked to increasing age and copies of the *MUC5B* genotype. The definite fibrosis pattern was associated with the highest risk of progression (odds ratio, 8.4). Probable UIP and UIP patterns predicted an increased risk of death compared to indeterminate UIP patterns. More specifically, patients with subpleural reticular changes, lower lobe predominant changes, or traction bronchiectasis had six times the probability of progression. All cases of honeycombing progressed over 5 years, while centrilobular nodules were a feature unlikely to progress. ILA progression increased the risk of death by 40% compared with those without ILA and almost doubled it (HR = 1.9) compared to those with ILA but without progression [11].

Two important studies [44,45] demonstrated that the presence of traction bronchiectasis/bronchiolectasis in ILA even in subjects without obvious fibrotic changes predicts worse survival. Moreover, the risk for shorter survival increases as traction bronchiectasis becomes more pronounced. Of note, subjects with ILA but without traction bronchiectasis/bronchiolectasis had similar overall survival compared to the group without ILA [44]. Traction bronchiectasis groups were associated with worse clinical outcomes, such as quality of life scores [45].

Hwang et al. [46] conducted a systematic review and meta-analysis to assess the radiological progression rate of interstitial lung abnormalities: the overall pooled progression rate was 47.1% (31.0% in follow-up <4.5 years and 64.2% in longer follow-up studies ≥4.5 years). Fibrotic ILAs were associated with a 5.55 times higher progression probability. The presence of reticulation is a major risk factor for progression in subpleural nonfibrotic ILA. In a large health check-up study of the Chinese population, the prevalence of ILA increased linearly with age, and 81% were characterized as subpleural nonfibrotic. Almost half of them progressed in 4 years, and the probability of progression increased with age, smoking habit, and higher degree of lung involvement. In fact, no difference was identified between subpleural nonfibrotic ILAs with extensive reticulation (OR = 4.4) and subpleural fibrotic ILAs (OR = 3.9) [9]. Finally, in a cohort of 1,384 individuals who underwent low-dose computed tomography screening for lung cancer [47], it was highlighted that a large portion of individuals (40.7%) were subsequently diagnosed with ILD; 53.8% had disease progression within 5 years, and 25.9% of individuals died. Fibrotic ILA predicted ILD diagnosis, disease progression, and mortality.

### 6.3. All-Cause Mortality

In a hallmark study, Putman et al. showed that ILAs are associated with higher all-cause mortality in four separate research cohorts (FHS (HR = 2.7), AGES-Reykjavik (HR = 1.3), COPDGene (HR = 1.8), and ECLIPSE (HR = 1.4)), and the associations were not attenuated after adjusting for smoking, cancer, COPD, and coronary artery disease [7]. In the AGES-Reykjavik cohort, the higher rate of mortality could be explained by a higher rate of death due to pulmonary fibrosis. However, overall, the contribution of ILAs to elevated mortality far exceeded, and could not be explained by, the progression to clinically relevant ILD. In a recent systematic review and meta-analysis of 22 studies with 88,325 participants, Grant-Orser et al. confirmed these observations of increased all-cause mortality and ILAs [21]. The authors estimated the prevalence and mortality risk of ILAs in three different types of populations: in lung cancer screening, in the general population, and in at-risk familial cohorts. In a median follow-up period of 5 years, the pooled mortality risk in the whole population was significantly higher in those with ILAs than those without (OR = 3.56). Specifically, ILAs were associated with increased risk of mortality in the lung cancer screening cohorts (OR = 2.41), in the general population cohorts (OR = 4.76), and in the familial cohort (OR = 5.81 in the one study available). Another recent systematic review also showed that ILAs significantly increased the risk of overall mortality (RR = 2.62) and lung cancer development (RR = 3.85) [48].

Could higher mortality in ILAs be explained by the multimorbidity often observed in these individuals? Sanders et al. [49] aimed to answer this question along with how mortality association with ILA compares to other age-related diseases, such as cardiovascular disease, diabetes mellitus, chronic kidney disease, chronic obstructive pulmonary disease, and cancer. In FHS, ILAs were associated with older age, greater pack-years of smoking, and a higher burden of cardiovascular disease, diabetes, chronic kidney disease, and cancer. In AGES-Reykjavik, ILAs were associated with older age, male sex, BMI, greater pack-years of smoking, and a higher burden of cardiovascular disease, chronic kidney disease, and COPD. The association of ILA with increased mortality is not explained by the prevalence of additional comorbidities, as the association between ILA and increased mortality remained after adjustment for the presence of all comorbidities. The presence of ILA increases mortality to at least an equal degree with these chronic diseases, and, after adjusting for shared confounders, the association of ILAs with the presence of other chronic diseases lacked consistency [49].

### 6.4. ILAs and COPD

The coexistence of ILAs affects the clinical course of COPD. Lee et al. [50] showed that definite ILAs were present in 30% of COPD patients; these individuals were older and had lower FEV1 and FVC compared to the non-ILA population. ILA presence was associated with an increased annual incidence of moderate to severe acute exacerbations and doubled the risk of frequent exacerbations. Additionally, the progression of ILAs was linked with increased annual decline in lung function [50]. A systematic review [51] examined the impact of coexistent ILAs on the outcomes of COPD or emphysema. ILAs were reported in 6.5% to 25.7% of the patients with COPD/emphysema, a prevalence higher than in the general population. COPD/emphysema patients with ILAs were older, mostly male, and had smoked more than those without ILAs. When ILAs were present, hospital admission and mortality rates were increased in COPD patients. FEV1 and FEV1% predicted tended to be higher in the group with ILAs without reaching significance in most studies. A recent study found elements of autoimmunity in COPD patients with ILA, as patients with ILA had positive antinuclear antibodies (ANA) more frequently compared to those without ILA, and positive ANA was linked with reduced diffusing capacity for carbon monoxide (DLCO) and a more pronounced restrictive pattern [52]. Finally, Zheng et al. attempted to elucidate possible associations between ILA and clinical features, as well as comorbidities, in 1,131 patients with COPD, of which 14.6% had concurrent ILAs [53]. The presence of ILA was not associated with demographic and clinical characteristics, nor specific laboratory parameters, except for levels of circulating fibrinogen and procalcitonin. Notably, in this study that included only COPD patients, the presence of ILA was inversely associated with the prevalence of lung cancer (OR = 0.50), particularly lung adenocarcinoma (OR = 0.32), a finding discrepant from studies in the general population. More importantly, ILAs were linked with extrapulmonary comorbidities such as heart failure (OR = 1.75) and cancers other than lung cancer (OR = 2.27) [53].

### 6.5. ILAs and Lung Cancer Risk and Risk Related to Oncologic Treatment Complications

The increased incidence of lung cancer among patients with IPF is well established. Axelsson et al. [54] showed that ILAs also are a risk factor for lung cancer in the general population. In this important study, there was a greater, approximately threefold, cumulative incidence of lung cancer diagnosis among participants with ILA than among participants without ILA, which remained robust after adjusting for age, sex, pack-years of smoking, and smoking. Both the presence and absence of a definite fibrosis imaging pattern increased the risk of lung cancer, although participants with definite fibrosis were at greater risk (adjusted HR = 3.95 for fibrotic ILAs vs. HR = 2.26 for nonfibrotic subtype) [54]. ILA increased the risk of a diagnosis of cancer overall, but this effect disappeared when lung cancers were excluded. Moreover, ILAs were associated with increased lung cancer mortality (HR = 2.89), and, again, the risk was much higher for the fibrotic subtype (HR = 5.98) compared to the nonfibrotic ILA (HR = 1.68) [54]. Whittaker Brown et al. [55] also showed that ILA is an independent risk factor for lung cancer. A total of 25,041 participants underwent low-dose CT imaging in the National Lung Screening Trial, and 20.2% had ILA. Those individuals had a higher incidence of lung cancer and higher disease-specific mortality. A recent systematic review and meta-analysis outlined the prognostic significance of interstitial lung abnormalities in lung cancer [56]. The pooled results from 12 studies showed that lung cancer patients with ILA had a reduced overall survival (HR = 2.22), reduced progression-free survival (HR = 1.59), and reduced cancer-specific survival (HR = 4.00) compared to those without ILA.

The presence of ILAs increases pulmonary complications after curative surgery for lung cancer by up to approximately 10 times. Prolonged air leakage, acute lung injury, pneumonia, and pneumothorax are the most commonly reported complications [57]. The 5-year overall survival rates are influenced too by the presence of ILAs (76% vs. 52% in the control group); when IPF is a comorbidity, the risk of complications rises to 56 times higher, and the survival rate falls down to 32% [58]. ILA was an independent risk factor for shorter survival in stage I non-small-cell lung cancer (NSCLC) in the Boston lung cancer study, where more pronounced ILA increased the risk for death (HR = 2.88) after adjusting for age, sex, smoking, and treatment [58]. Similarly, ILA on baseline CT at diagnosis of stage IV NSCLC was a predictor of poor clinical outcomes, as it was associated with shorter overall survival (HR = 2.09) [59].

Pre-existing interstitial lung abnormalities are risk factors for immune checkpoint inhibitor-induced interstitial lung disease in non-small cell lung cancer and ground glass attenuation pattern is an independent risk factor [60]. This increased risk was also demonstrated when immune checkpoint inhibitors were used in other types of cancer as well, whereas a 6-fold probability of drug-induced ILD was observed [61]. Additionally, cancer patients with ILA have a more than doubled risk of developing significant radiation pneumonitis [48]. These observations suggest an important role for the presence of ILAs in the outcomes of lung cancer patients after treatment.

Table 2 summarizes ILA-associated risks in relation to the radiological subtype.

## 7. How Should ILAs Be Monitored?

The increasing use and better quality of CT scans along with growing evidence of the clinical significance of ILAs put physicians in the difficult position of deciding how patients with ILAs should be monitored. Considering the fact that the systematic evaluation of large cohorts has shown that ILAs are common and that the natural history of ILAs is variable, effective screening and appropriate monitoring are challenging as appropriate follow-up strategies are not available. Prompt diagnosis and management is of particular significance, focusing on personalized risk factor modification and treatment, if needed, in order to interrupt the vicious journey toward poor outcomes (Figure 2 and Figure 3).

In an attempt to construct consensus-based expert recommendations to assist clinicians in the recognition, referral, and follow-up of patients with ILA, an expert survey initiative recruited pulmonologists and radiologists with expertise in ILD [62]. Overall, 44 experts participated, and consensus was reached on 39 of 85 questions (the threshold for consensus was defined a priori as 75% agreement or disagreement). Regarding ILAs observed during lung cancer screening, it was agreed that the presence of honeycombing and traction bronchiectasis or bronchiolectasis is indicative of potentially progressive ILD. Moreover, incident honeycombing should be reported by the radiologist as a potential recommendation for referral to a pulmonologist. Expert opinions regarding asymptomatic patients were that if non-dependent subpleural reticulation, traction bronchiectasis or honeycombing, or centrilobular ground-glass nodules or patchy ground-glass opacity are observed on CT imaging, HRCT imaging (89%, 84%, and 75% agreement, respectively) and full pulmonary function tests (PFTs) should be ordered (98%, 95%, and 75% agreement, respectively). Patients with honeycombing or traction bronchiectasis should be referred to a pulmonologist irrespective of PFT. For patients with ILA but normal PFT, 90% of expert panel members recommended follow-up, specialist referral, or periodic testing with full PFTs. No agreement was achieved on whether HRCT scanning should be additionally performed. Regarding ILA screening in high-risk populations, the majority agreed that patients with systemic sclerosis >50 years old should be screened for ILD. In patients with rheumatoid arthritis, when crackles are absent on lung auscultation, consensus on screening was not reached (64% agreed). Finally, in high-risk populations that have more than one relative with ILD, substantial agreement (73%) occurred for screening, even in patients without crackles. Consensus was reached only on the use of full PFTs, not HRCT.

Several artificial intelligence methods have been proposed for the automatic identification of ILD patterns [63] and may be of particular help in the identification and progression risk assessment of ILAs. A novel methodology for automated identification and classification of ILA patterns in CT images using deep convolutional neural networks was described by Bermejo-Peláez et al. [64]. The researchers trained and tested the system using multiple radiographic tissue samples corresponding to eight different parenchymal feature classes (normal parenchyma, ground-glass, reticular, nodular, linear, subpleural line, paraseptal emphysema, and centrilobular emphysema) and then incorporated two-and three-dimensional architectures. The system was able to identify and classify ILA with an average sensitivity of 91.4% and an average specificity of 98.18% [64]. Quantification of pulmonary fibrosis calculated by computer-aided detection at preoperative CT in patients with lung cancer was an independent predictor of disease-free survival [65]. A recent very interesting study [66], using deep learning-based ILA quantification, found that the median time to radiologic progression of ILA was 3.2 years. The % extent of fibrotic ILA in the whole lung was an independent risk factor for both ILA progression and progression to UIP, and although a 3-year interval follow-up after the detection of ILA might be appropriate, the authors concluded that patients with extensive fibrosis and the presence of honeycombing may benefit from shorter intervals [66].

A simple proposed algorithm is shown in Figure 4. As a first step, individuals identified with ILA should be investigated on whether they belong to high-risk populations; those with familiar fibrosis history, CTD history, or relevant clinical and/or laboratory results or occupational and/or domestic exposure to metal and/or organic substances need appropriate identification and management. Subsequently, thorough exploration for possible relevant symptoms and lung function tests should follow to capture individuals with mild ILD. After physiologically impaired subjects are excluded, the most difficult part is to stratify ILAs with a high likelihood of progression to clinically relevant ILD versus ILAs with a small risk of this happening. Imaging profiles are very helpful, with special attention to UIP patterns, subpleural fibrosis, and the presence of extensive reticulation. Risk factors should be eliminated in all groups, and closer follow-up, when appropriate, will permit treating ILDs in the early phases and possibly modify the course of the disease. Additionally, physicians should be alert and regard ILAs as an important comorbidity with special caution in patients undergoing thoracic surgery or other therapies in the application of positive pressure ventilation (due to the increased risk of acute lung injury) [10], while medications known to cause ILD should be avoided if possible.

## 8. Conclusions and Future Directions

ILA is a relatively novel entity with significant morbidity and mortality implications for the high-risk and general older adult populations, which every clinician should have in mind. Radiology reports should not omit reporting even subtle changes, especially in scans with concomitant pathology, and it is essential that findings are described in detail in terms of morphology and distribution, as different implications are encompassed. New technologies, such as artificial intelligence and deep learning, will definitely be the next step in mapping ILAs objectively according to progression and mortality risk. A more holistic approach would include the identification of biomarkers able to predict ILA progression. The ultimate goal remains managing ILDs early in the course of the disease in order to improve unfavorable outcomes. Early detection and management of ILA will serve this purpose. Future studies should focus on a better understanding of the pathogenesis and on drawing widely available, effective treatment strategies for ILA.

## Figures and Tables

**Figure 1 diagnostics-15-00509-f001:**
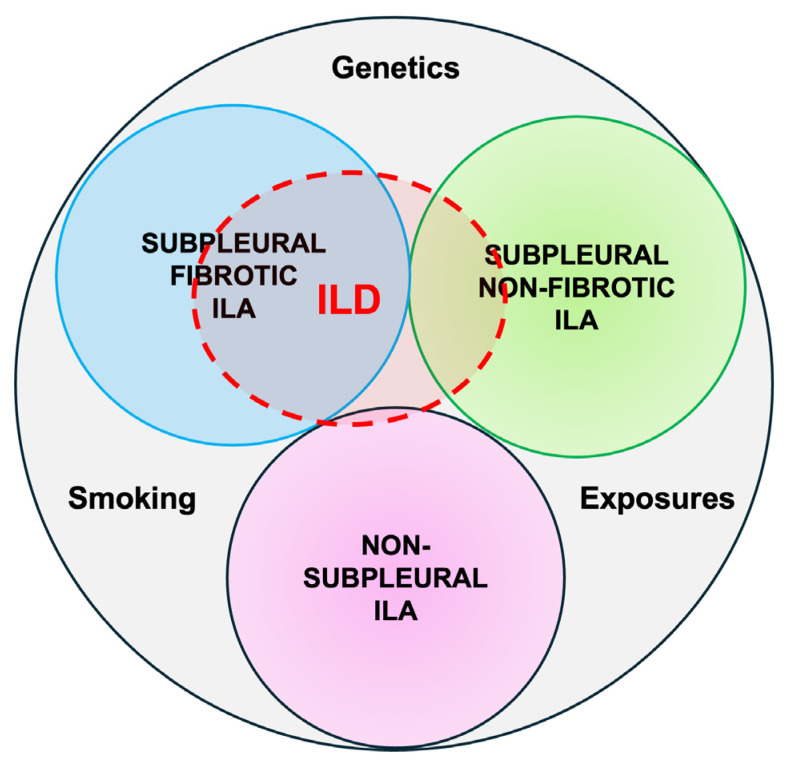
Radiological types of ILAs.

**Figure 2 diagnostics-15-00509-f002:**
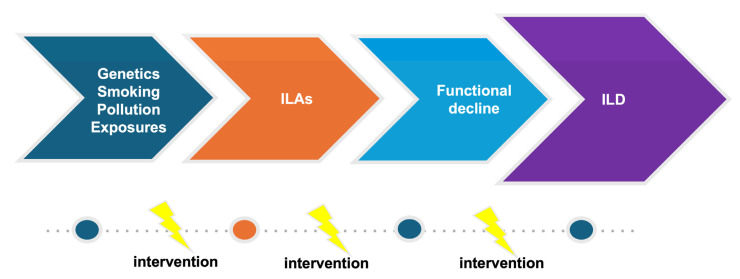
Evolution of ILD.

**Figure 3 diagnostics-15-00509-f003:**
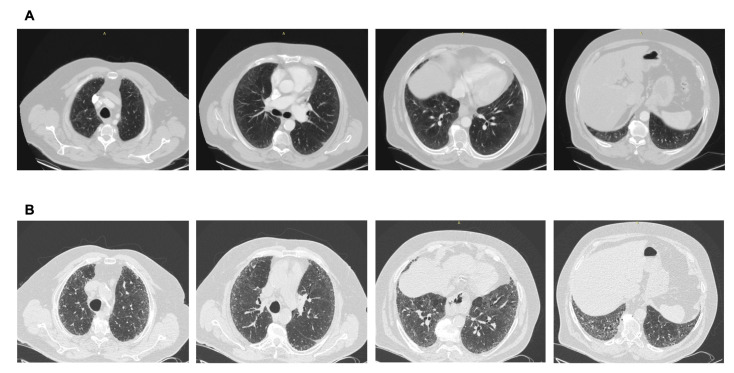
(**A**) CT scan performed after a car accident in an asymptomatic individual. Note the mild ground glass subpleural changes and the subtle mosaic attenuation. No investigation/intervention followed. (**B**) Three years later, the patient became symptomatic, with remarkable progression to a fibrotic pattern. A heavy exposure to pigeons was revealed. Final diagnosis: Hypersensitivity pneumonitis. Avoidance of the causative factor would possibly have attenuated progression.

**Figure 4 diagnostics-15-00509-f004:**
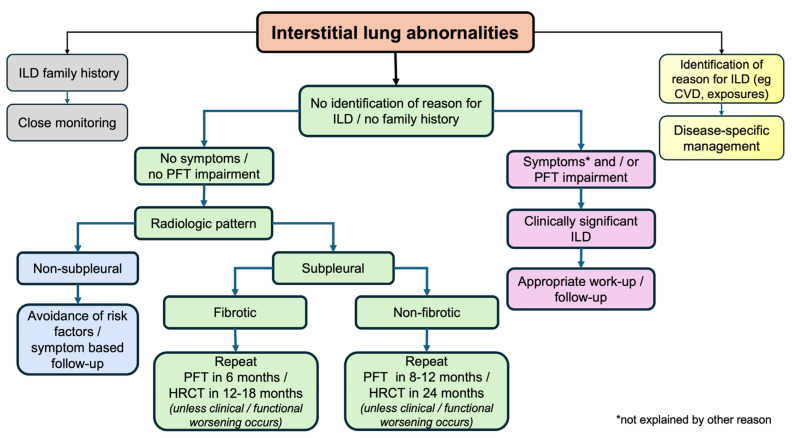
A simple algorithm for the management of ILAs. CVD: collagen vascular disease; HRCT: high-resolution computed tomography; ILD: interstitial lung disease; PFT: pulmonary function tests.

**Table 1 diagnostics-15-00509-t001:** What is and what is not ILA.

**ILA** *involving at least 5% of a lung zone, NOT suspected ILD*		**NOT ILA**
**Subpleural distribution** *Likely to progress*	**Non-subpleural distribution** *Unlikely to progress*	
Ground-glass Reticular abnormalities Non-emphysematous cysts Honeycombing Traction bronchiectasis/bronchiolectasis		Dependent lung atelectasis Focal paraspinal fibrosis Mild focal or unilateral abnormality Interstitial oedema Pleuropulmonary fibroelastosis Findings of aspiration Smoking-related centrilobular nodularity

**Table 2 diagnostics-15-00509-t002:** ILA subtypes and their associated risks.

ILA Subtype	Risk of Progression	Risk of Mortality	Risk of Lung Cancer
**Fibrotic ILAs**			
Fibrotic ILAs	OR 5.55 [46]	HR 3.8 [47]	HR 3.95 (incidence) [54] HR 5.98 (mortality) [54]
Definite fibrosis	OR 8.4 [11]		
Subpleural fibrotic ILAs	OR 3.9 [9]		
Traction bronchiectasis: scores 1, 2, and 3	57%, 90%, and 100% patients progressed, respectively [44]	HR 2.18, 2.65, and 6.8, respectively [44]	
Probable UIP		OR 1.9 [11]	
Definite UIP		OR 4.5 [11]	
**Nonfibrotic ILAs**			
Nonfibrotic ILAs			HR 2.26 (incidence) [54] HR 1.68 (mortality) [54]
Subpleural reticulation	OR 6.6 [11]	OR 1.6 [11]	
Subpleural reticulation	OR 1.9 [9]		
Extensive reticulation	OR 4.4 [9]		
No traction bronchiectasis	10% patients progressed [44]		

ILA: interstitial lung abnormality; UIP: usual interstitial pneumonia pattern; OR: odds ratio; HR: hazard ratio.

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
