# Peer review of "Interstitial Lung Abnormalities: Unraveling the Journey from Incidental Discovery to Clinical Significance"

_diagnostics, 2025, doi:10.3390/diagnostics15040509_

Round 1

Reviewer 1 Report

Comments and Suggestions for Authors

My Evaluation Report

1. Introduction

Evaluation:

The introduction section adequately addresses the definition and importance of interstitial lung abnormalities (ILA). However, it lacks a clear statement on the gaps in the literature and the specific contributions of this review.

 Recommendation:  Include a statement highlighting the gaps in the literature. For example:

“While the clinical significance of ILA is increasingly recognized, there remains a lack of comprehensive reviews addressing its association with cancer, COPD, and genetic predispositions.”

2. ILA and Cancer

Evaluation: The relationship between ILA and lung cancer has been addressed. However, the comparison between fibrotic and non-fibrotic ILA subtypes in terms of cancer risk is superficial.

Recommendation:

Cite additional studies for a more detailed analysis, such as:

"The associations of interstitial lung abnormalities with cancer diagnoses and mortality.", DOI: 10.1183/13993003.02154-2019

3. ILA and COPD

Evaluation: The connection between ILA and COPD is discussed. However, the clinical outcomes and progression of ILA in COPD patients need more detail.

Recommendation:

Please include a reference to recent findings:

“The prevalence, risk factors, and clinical implications of ILA in COPD patients are comprehensively analyzed in Zheng et al. (2024).” (BMC Pulmonary Medicine)

4. ILA and Genetics

Evaluation:  Genetic predisposition, including the MUC5B promoter polymorphism, is addressed. However, the clinical implications of genetic factors in ILA are not fully explored.

Recommendation: please add references to studies such as: 

"Genetic and environmental factors in interstitial lung diseases: current and future perspectives on early diagnosis of high-risk cohorts.", DOI: 10.3389/fmed.2023.1232655

5. ILA and Environmental Exposures

Evaluation: The effects of environmental and occupational exposures on ILA are discussed. However, there is limited analysis of air pollution and smoking.

 Recommendation:

 Include references to key studies, for instance: “The impact of air pollution and smoking on ILA development is highlighted in Grant-Orser et al. (2023), originally published in American Journal of Respiratory and Critical Care Medicine.” (ATS Journal)

6. ILA and All-Cause Mortality

Evaluation: The relationship between ILA and all-cause mortality is addressed, but additional references to recent meta-analyses could strengthen this section.

 Recommendation:  Add studies like:

“The relationship between ILA and all-cause mortality is thoroughly explored by Grant-Orser et al. (2023).” (ATS Journal)  

7. Use of Figures and Tables

Evaluation:  The use of visual materials is limited. Adding more tables or diagrams summarizing ILA subtypes and associated risk factors would enhance clarity.

Recommendation:

Please include comparative  tables for ILA subtypes and their associated risks.

8. Conclusion

Evaluation:

The conclusion summarizes the findings but does not sufficiently address the clinical implications and future research directions.

Recommendation:

Expand the conclusion to include statements such as:

“Early detection and management of ILA could improve patient outcomes. Future studies should focus on the pathogenesis and treatment strategies of ILA.”

General Comments

If no directly matching studies exist, I encourage the authors to explore similar research in the field to strengthen their review.

Strengths:  Comprehensive coverage of ILA’s clinical significance. Logical structure with relevant subtopics. Weaknesses:  Limited references to recent meta-analyses and systematic reviews.

Insufficient use of visual materials.  

Recommendation:   Minor revisions are suggested to enhance the review's impact.

Comments on the Quality of English Language

5. Language, Clarity, and Originality

  • Language Quality: The manuscript is generally well-written, but certain sentences are overly long and complex.

Suggestion: "I think that simplifying lengthy and complex sentences may improve readability and flow."

Author Response

Comment 1. Introduction

Evaluation: The introduction section adequately addresses the definition and importance of interstitial lung abnormalities (ILA). However, it lacks a clear statement on the gaps in the literature and the specific contributions of this review.

 Recommendation:  Include a statement highlighting the gaps in the literature. For example: “While the clinical significance of ILA is increasingly recognized, there remains a lack of comprehensive reviews addressing its association with cancer, COPD, and genetic predispositions.”

Response 1

Thank you for pont this out. We agree, therefore we have made the following changes in the introduction (page 2, lines 50-53):

“The present review aims to summarize the most up to date knowledge on ILAs and highlight the need to identify patients at greatest risk of progression to ILD and worse clinical outcomes, with the final goal to eliminate potential risk factors and offer appropriate early treatment. Besides contributing to disease awareness, we aimed to comment in detail on issues not adequately addressed such as the association between ILAs and genetic predispositions, cancer and COPD and also provide a simple framework of the management of ILAs for the practicing clinician.”

Comment 2

ILA and Cancer

Evaluation: The relationship between ILA and lung cancer has been addressed. However, the comparison between fibrotic and non-fibrotic ILA subtypes in terms of cancer risk is superficial.

Recommendation: Cite additional studies for a more detailed analysis, such as: "The associations of interstitial lung abnormalities with cancer diagnoses and mortality."DOI: 10.1183/13993003.02154-2019

Response 2

We agree with the reviewer. We had already included this reference (new reference 55), and enriched the document underlining the differences between the ILA subtypes, using information from the suggested study (page 9, lines 369-373).

“Both presence and absence of a definite fibrosis imaging pattern increased the risk of lung cancer, although participants with definite fibrosis were at greater risk (adjusted HR=3.95 for fibrotic ILAs vs HR=2.26 for non-fibrotic subtype) [55]. ILA increased the risk of diagnosis of cancer overall but this effect disappeared when lung cancers were excluded. Moreover, ILAs were associated with increased lung cancer mortality (HR = 2.89 and again the risk was much higher for the fibrotic subtype (HR 5.98) compared to the non-fibrotic ILA (HR 1.68) [55].”

Comment 3. ILA and COPD

Evaluation: The connection between ILA and COPD is discussed. However, the clinical outcomes and progression of ILA in COPD patients need more detail.

Recommendation:

Please include a reference to recent findings: “The prevalence, risk factors, and clinical implications of ILA in COPD patients are comprehensively analyzed in Zheng et al. (2024).” (BMC Pulmonary Medicine)

Response 3

We thank the reviewer for the suggestion. We have included the study of Zheng et al. and enriched the manuscript as follows (page 9, lines 351-360):

“Finally, Zheng et al attempted to elucidate possible associations between ILA and clinical features, as well as co-morbidities in 1131 patients with COPD, of which 14.6% had concurrent ILAs [54]. The presence of ILA was not associated with demographic and clinical characteristics, nor specific laboratory parameters, except for levels of circulating fibrinogen and procalcitonin. Notably, in this study that included only COPD patients, the presence of ILA was inversely associated with the prevalence of lung cancer (OR = 0.50), particularly lung adenocarcinoma (OR 0.32), a finding discrepant from studies in the general population. More importantly, ILAs were linked with extrapulmonary comorbidities such as heart failure (OR = 1.75) and cancers other than lung cancer (OR = 2.27) [54].

Comment 4. ILA and Genetics

Evaluation: Genetic predisposition, including the MUC5B promoter polymorphism, is addressed. However, the clinical implications of genetic factors in ILA are not fully explored.

Recommendation: please add references to studies such as:

"Genetic and environmental factors in interstitial lung diseases: current and future perspectives on early diagnosis of high-risk cohorts."DOI: 10.3389/fmed.2023.1232655

Response 4

We thank the reviewer for the comment. We agree and therefore we have modified the manuscript as follows, including the suggested reference (page 6, lines 213-217):

“The clinical implications of genetic factors in ILA are more obvious in relatives of familial pulmonary fibrosis patients, in whom ILAs on chest CT scans are common, with a reported prevalence in first-degree relatives of 14–25%. Among them, 63.3% of patients with limited ILAs at baseline experienced progression, while 19.4% developed extensive HRCT abnormalities or clinical ILD at 5 years [38].

Comment 5. ILA and Environmental Exposures

Evaluation: The effects of environmental and occupational exposures on ILA are discussed. However, there is limited analysis of air pollution and smoking.

 Recommendation:

 Include references to key studies, for instance: “The impact of air pollution and smoking on ILA development is highlighted in Grant-Orser et al. (2023), originally published in American Journal of Respiratory and Critical Care Medicine.” (ATS Journal)

Response 5

We thank the reviewer for his points. The proposed reference has already been referenced in our paper. We also added some additional details relevant to smoking and air pollution as follows (page 5, lines 160-161):

“Air pollution also merits special attention, as it seems to be involved in pathogenetic pathways as oxidative stress, inflammation, and telomere shortening [26]”

And (page 5, lines 153-159):

“The specific ILA mostly associated with current smoking is the pattern of centrilobular nodules (OR, 4.82) [12]. In a cohort of 352 patients that underwent resection of lung nodules or masses, those with radiographic ILAs/ILDs were more frequently ever smokers (49% vs. 39.9% with no ILAs), with more pack years of smoking (44.57 ± 36.21 vs. 34.96 ± 26.22), more often with concurrent COPD (35% vs. 26.5%) and radiographic evidence of centrilobular (40% vs. 22.2%) and paraseptal emphysema (17.6% vs. 9.6%) [25].”

And (page 5, lines 172-176):

“By contrast, air pollution exposure was not an independent risk factor for ILA occurrence in the SPIROMICS population with COPD. Association between pollution and ILA was affected by the presence of genetic polymorphisms (RR per 26 ppb NOx, 2.41 and RR per 4 μg/m3 PM2.5, 1.43, for curriers) and the history of smoking habit (RR per 10 ppb, NO2 1.64 for former smokers) [29].”

Comment 6. ILA and All-Cause Mortality

Evaluation: The relationship between ILA and all-cause mortality is addressed, but additional references to recent meta-analyses could strengthen this section.

 Recommendation: Add studies like: “The relationship between ILA and all-cause mortality is thoroughly explored by Grant-Orser et al. (2023).” (ATS Journal)

Response 6

Thank you for pointing this out. We had already mentioned that study in the section, but, indeed it deserved a more detailed description. Therefore we have changed the phrase “A subsequent meta-analysis confirmed these observations of increased all-cause mortality in the general population (OR 4.76) and in lung cancer screening cohorts (OR 2.4) [21]”, as follows (page 8, lines 311-319):

“In a recent systematic review and meta-analysis of 22 studies with 88,325 participants, Grant-Orser et al confirmed these observations of increased all-cause mortality and ILAs [22]. The authors estimated the prevalence and the mortality risk of ILAs in 3 different types of population: in lung cancer screening, in general population, and at-risk familial cohorts. In a median follow-up period of 5 years. The pooled mortality risk in the whole population, was significantly higher in those with ILAs than those without (OR 3.56). Specifically, ILAs were associated with increased risk of mortality in the lung cancer screening cohorts (OR, 2.41), in the general population cohorts (OR, 4.76), and in the familial cohort (OR, 5.81 in the one study available). Another recent systematic review also showed that ILAs significantly increase risk of overall mortality (RR 2.62) and lung cancer development (RR 3.85) [49].”

Comment 7. Use of Figures and Tables

Evaluation: The use of visual materials is limited. Adding more tables or diagrams summarizing ILA subtypes and associated risk factors would enhance clarity.

Recommendation:

Please include comparative tables for ILA subtypes and their associated risks.

Response 7

Following your advice, we have improved the quality of the existing images. We have also included a table with ILA subtypes and their associated risks (Table 2). We have also added the sentence in page 10, line 401: Table 2 summarizes ILA associated risks in relation to the radiological subtype.

Table 2. ILA subtypes and their associated risks

ILA subtype

Risk of progression

Risk of mortality

Risk of lung cancer

Fibrotic ILAs

Fibrotic ILAs

OR 5.55 [47]

HR 3.8 [48]

HR 3.95 (incidence) [55]

HR 5.98 (mortality) [55]

Definite fibrosis

OR 8.4 [11] 

Subpleural fibrotic ILAs

OR 3.9 [9]

Traction bronchiectasis:

score 1, 2, 3

% patients progressed 57%, 90%, and 100%, respectively [45]

HR 2.18, 2.65, and 6.8, respectively [45]

Probable UIP

OR 1.9 [11]

Definite UIP

OR 4.5 [11]

Nonfibrotic ILAs

Nonfibrotic ILAs

HR 2.26 (incidence) [55]

HR 1.68 (mortality) [55]

Subpleural reticulation

OR 6.6 [11]

OR 1.6 [11]

Subpleural reticulation

OR 1.9 [9]

Extensive reticulation

OR 4.4 [9]

No traction bronchiectasis

10% patients progressed [45]

ILA: interstitial lung abnormality; UIP: usual interstitial pneumonia pattern;, OR: odds ratio; HR: hazard ratio.

Comment 8. Conclusion

Evaluation:

The conclusion summarizes the findings but does not sufficiently address the clinical implications and future research directions.

Recommendation: Expand the conclusion to include statements such as:

“Early detection and management of ILA could improve patient outcomes. Future studies should focus on the pathogenesis and treatment strategies of ILA.”

Response 8

Thank you, we have added in the conclusion the phrase (page 13, lines 496-498):

"Early detection and management of ILA will serve this purpose. Future studies should focus on better understanding of the pathogenesis and on drawing widely available, effective treatment strategies of ILA.”

General Comments

If no directly matching studies exist, I encourage the authors to explore similar research in the field to strengthen their review.

Response: Thank you fort his suggestion. We have provided comments at several places of our review to address this.

Strengths: Comprehensive coverage of ILA’s clinical significance. Logical structure with relevant subtopics. Weaknesses: Limited references to recent meta-analyses and systematic reviews. Insufficient use of visual materials.

Response: We have followed the reviewer's advice and we have included all the available references in the field to the best of our knowledge. We have improved the quality of our 4 Figures and we have also added an additional Table, according to the reviewer's suggestions.

Recommendation: Minor revisions are suggested to enhance the review's impact.

Comments on the Quality of English Language

  1. Language, Clarity, and Originality
  • Language Quality: The manuscript is generally well-written, but certain sentences are overly long and complex.

Suggestion: "I think that simplifying lengthy and complex sentences may improve readability and flow."

Response: Thank you for your advice. We reviewed the manuscript and simplified most of the complex sentences. Changes are highlighted throughout the text.

Reviewer 2 Report

Comments and Suggestions for Authors

The review "Interstitial lung abnormalities: unraveling the journey from incidental discovery to clinical significance is very relevant and is well written. The review is well organized and presents good information to the clinicians. I suggest that figure 4 can be amplified to a better visualization .

Author Response

Comment: The review "Interstitial lung abnormalities: unraveling the journey from incidental discovery to clinical significance is very relevant and is well written. The review is well organized and presents good information to the clinicians. I suggest that figure 4 can be amplified to a better visualization

Response to Comment: Thank you for your kind comments. We have now adapted and amplified Figure 4 for better visualization. 

Reviewer 3 Report

Comments and Suggestions for Authors

The main question of “Interstitial lung abnormalities: unraveling the journey from incidental discovery to clinical significance” paper by Athena Gogali, Christos Kyriakopoulos, and Konstantinos Kostikas is a detailed description of currently published findings on interstitial lung abnormalities (ILAs). While ILA is a quite novel radiological entity, it has a great importance for clinicians. The authors fairly noted that the importance of ILAs has risen due to the increasing prevalence of interstitial lung diseases (ILD) and the need to identify patients with idiopathic pulmonary fibrosis and progressive pulmonary fibrosis to start antifibrotic therapy as early as possible. This paper is useful for clinicians as it provides the algorithm on the management of patients with different variants of ILA and identification of ILA with the potential to progress.

The paper contains one table and 4 figures.

The conclusion made by the authors is consistent with the study goals.

The references are appropriate to the topic and the content of the paper.

Author Response

Comments: 

The main question of “Interstitial lung abnormalities: unraveling the journey from incidental discovery to clinical significance” paper by Athena Gogali, Christos Kyriakopoulos, and Konstantinos Kostikas is a detailed description of currently published findings on interstitial lung abnormalities (ILAs). While ILA is a quite novel radiological entity, it has a great importance for clinicians. The authors fairly noted that the importance of ILAs has risen due to the increasing prevalence of interstitial lung diseases (ILD) and the need to identify patients with idiopathic pulmonary fibrosis and progressive pulmonary fibrosis to start antifibrotic therapy as early as possible. This paper is useful for clinicians as it provides the algorithm on the management of patients with different variants of ILA and identification of ILA with the potential to progress.

The paper contains one table and 4 figures.

The conclusion made by the authors is consistent with the study goals.

The references are appropriate to the topic and the content of the paper.

Response: Thank you very much for your kind comments

Round 2

Reviewer 1 Report

Comments and Suggestions for Authors

      I appreciate the authors' efforts in addressing the concerns raised in the previous review. The manuscript has undergone significant improvements, and many key areas have been refined to enhance clarity and scientific rigor. Below, I summarize the most notable improvements:

Key Improvements

Introduction and Background

  • The distinction between Interstitial Lung Abnormalities (ILAs) and Interstitial Lung Disease (ILD) has been clarified, making the clinical relevance of ILAs more apparent.
  • Additional references have been included, strengthening the discussion on the prevalence and prognostic value of ILAs.
  • Definitions and Classification
  • The manuscript now adheres more closely to the Fleischner Society’s 2020 guidelines, improving consistency and reducing ambiguity.
  • The differentiation between subpleural and non-subpleural ILAs has been better structured, enhancing its clinical applicability.

Genetics and Risk Factors

  • The MUC5B promotor variant and other genetic predispositions have been discussed in greater depth, supported by more relevant literature.
  • The role of environmental and occupational exposures, smoking, and aging has been expanded for a more comprehensive understanding.

Association with Cancer and COPD

  • The link between ILAs and lung cancer progression has been strengthened with additional supporting studies.
  • The connection between ILAs and COPD pathophysiology is now more clearly articulated, aligning with the latest research.

Clinical Approach and Management

  • The monitoring and management strategies for ILAs, which were previously lacking, have now been outlined in a more practical way.
  • A structured approach for clinicians has been introduced, improving the manuscript’s usefulness for real-world application.